# SALpingectomy for STERilisation (SALSTER): study protocol for a Swedish multicentre register-based randomised controlled trial

Leonidas Magarakis ![ORCID],[1] Annika Idahl,[2] Karin Sundfeldt,[1,3] Per Liv,[4] Mathias Pålsson,[1] Annika Strandell[1,3]

¹Department of Obstetrics and Gynecology, Institute of Clinical Sciences, Sahlgrenska Academy, University of Gothenburg, Gothenburg, Sweden
²Department of Clinical Sciences, Obstetrics and Gynecology, Umeå University, Umeå, Sweden
³Department of Obstetrics and Gynecology, Sahlgrenska University Hospital, Gothenburg, Sweden
⁴Department of Public Health and Clinical Medicine, Umeå University, Umeå, Sweden

**Correspondence to**
Dr Leonidas Magarakis;
leonidas.magarakis@gu.se

## ABSTRACT

**Introduction** Salpingectomy is currently suggested as an alternative to tubal ligation for sterilisation. Precursor lesions of ovarian carcinoma can be found in the fallopian tubes; thus, salpingectomy could possibly reduce the incidence. Most of the existing trials on safety are small, on caesarean section and report on surrogate ovarian function measures. Randomised trials in laparoscopy are lacking. Well-designed trials are needed to evaluate safety of laparoscopic opportunistic salpingectomy.

**Methods and analysis** In SALSTER, a national register-based randomised controlled non-inferiority trial, 968 women <50 years, wishing laparoscopic sterilisation will be randomised to either salpingectomy or tubal ligation. The Swedish National Quality Register of Gynecological Surgery (GynOp) will be used for inclusion, randomisation and follow-up. Primary outcomes are *any complication* up to 8 weeks postoperatively, and *age at menopause*. Both outcomes are measured with questionnaires, complications are also assessed by a gynaecologist. In a nested trial, ovarian function will be evaluated comparing the mean difference of anti-Müllerian hormone (AMH), assessed preoperatively and 1 year after surgery.

**Ethics and dissemination** Performing salpingectomy for sterilisation has become increasingly common, despite the unclear risk-benefit balance. SALSTER studies the safety of salpingectomy compared with tubal ligation. Regardless of the result, SALSTER will provide gynaecologists with high quality evidence to inform women to decide on salpingectomy or not. The central ethical review board of Gothenburg, Sweden (Dnr. 316–18) approved the trial in 2018. Results will be presented at scientific congresses and published in peer-reviewed scientific journals. The results will be communicated through professional organisations and research networks.

**Trial registration number** NCT03860805.

## STRENGTHS AND LIMITATIONS OF THIS STUDY

⇒ The register-based randomised controlled trial combines the advantages of two study designs: the randomised trial with unbiased allocation to minimise confounding and the observational register study with an automated and cost-efficient follow-up.

⇒ Using the GynOp (Swedish National Quality Register of Gynecological Surgery) register as a platform allows all trial components (identification of eligible patients, communication regarding study information and giving informed consent, randomisation and follow-up questionnaires) to be conducted within the register.

⇒ The use of the Swedish personal identification number allows cross-linking of the study cohort with multiple registers for the long-term follow-up.

⇒ The multicentre design enhances the generalisability of the results.

⇒ The nature of the trial makes blinding of the patients very difficult and impossible for the surgeons.

## INTRODUCTION

The use of salpingectomy as a sterilisation procedure is increasing, due to the theory of high-grade serous ovarian carcinoma (HGSC) originating from the fallopian tube. Epithelial ovarian cancer (EOC) is a group of heterogeneous malignancies regarding origin, molecular biology, morphology, gene expression and clinical behaviour. Precancerous lesions, serous tubal intraepithelial carcinomas (STIC), detected in the tubal epithelium are suggested to be the origin of EOC, particularly HGSC. Dysplastic cells may shed from STIC lesions and implant on the ovaries and/or peritoneum and develop into HGSC.[1] Opportunistic salpingectomy to remove the potential site of origin as a preventive measure is therefore suggested for women who wish permanent sterilisation.[2 3]

Tubal ligation is by itself associated with some protection against EOC.[4] Fallopian tubes may act as a conduit of either malignant or normal cells from the endometrial cavity to the ovaries. These cells may give rise to endometrioid and clear-cell carcinomas directly or indirectly by malignant transformation of benign conditions such as endometriosis.[5] Possibly, salpingectomy could add

to the protective effect of tubal ligation by removing the fimbriated end of the fallopian tubes where STIC lesions may develop.[4 6 7]

Several gynaecological societies recommend physicians to inform women planned to undergo sterilisation, that bilateral salpingectomy instead of tubal ligation, is an option.[2 3] This recommendation is based on observational studies showing that *indicated* salpingectomy compared with no surgery, is associated with a decreased EOC incidence.[4 6 7] The effect size of *opportunistic* salpingectomy compared with tubal ligation is unknown.

There are safety concerns, since salpingectomy increases surgical trauma compared with tubal ligation. This may increase perioperative complications and may also affect blood and nerve supply to the ovaries, impairing ovarian function and possibly, in the long-term, cause an earlier menopause.[8] Systematic reviews comparing salpingectomy with tubal ligation for safety outcomes such as reoperation, intraoperative complications, blood loss, wound infections, etc, have identified studies with various limitations.[9] All published randomised controlled trials (RCTs) are small and conducted at caesarean section. They report on surrogate measures of endocrine function and demonstrate no difference in the short-term.[10–12] Many of the published cohort studies are small and underpowered to study complications. Sterilisation is more commonly performed by laparoscopy, especially after the hysteroscopic salpingeal occluding technique with permanent implants was withdrawn from the market due to adverse effects.[13] No trial has reported on the outcome EOC. A large retrospective cohort study detected no difference in time to menopausal symptoms when comparing women who had undergone salpingectomy or tubal ligation. However, the follow-up period was insufficiently short to analyse menopausal symptoms.[14] Well-designed randomised trials of laparoscopic sterilisation procedures are needed to compare salpingectomy with tubal ligation regarding both surgical outcomes and clinical endpoints of ovarian function.

This register-based randomised trial will study the safety of laparoscopic salpingectomy for sterilisation compared with tubal ligation. The specific aim is to analyse if the risk of complications and hormonal side effects do not increase beyond predefined non-inferiority margins after salpingectomy compared with tubal ligation.

## METHODS AND ANALYSIS
### General study design

SALSTER, a national register-based, RCT will compare two laparoscopic procedures for sterilisation: salpingectomy and tubal ligation for safety aspects, in women without known hereditary risk for EOC.

In the primary analyses, SALSTER will test the hypotheses that salpingectomy compared with tubal ligation for laparoscopic sterilisation:

▶ Does not increase the risk for complications perioperatively and up to 8 weeks postoperatively.
▶ Does not cause earlier menopause, assessed as age at onset of natural menopause.

### The GynOp register

The SALSTER trial is conducted within the Swedish National Quality Register of Gynecological Surgery (GynOp).[15] GynOp is used by all gynaecological departments in Sweden. Inclusion and participation in national quality registers in Sweden is regulated by law[16]; patients are informed of their inclusion in the register, with an 'opt-out' clause which, if activated, enables the patient to have all personal data removed from the register. The GynOp database is approved for use by healthcare systems under the supervision of the Swedish Data Protection Authority. All information is stored on secured servers at Region Västerbotten. Background health data, information on surgical procedures, diagnoses, complications at 8 weeks and 1 year postoperatively are routinely recorded in GynOp. Women planning for gynaecological surgery receive a personal password that allows them to logon to GynOp to answer preoperative and follow-up questionnaires. Data input in GynOp is mainly web-based, but printouts of questionnaires can be used if needed. The data collection forms and questionnaires are available from www.gynop.org on request.

All gynaecological departments reporting data to the register received information about the trial and were automatically included unless a department actively declined participation. A list of gynaecological departments participating in the study can be provided by the GynOp office in Umeå on demand. Both regional and academic gynaecological departments are participating in the study. The Swedish network for National Clinical Studies in Obstetrics and Gynecology (SNAKS) is actively involved and improves collaboration between healthcare providers engaged in the trial.[17]

A specific SALSTER application has been added to GynOp to complement existing routines. This module includes screening of eligibility, presentation of study information and opportunity to give informed consent online, as well as randomisation and trial-specific questionnaires preoperatively and for follow-up.

Preoperatively, basic baseline demographic variables are registered routinely. Added to these variables are questions on menstruation pattern, age at menarche, duration of breast feeding, previous and present use of hormonal contraceptives and previous Chlamydia infection or salpingitis to assess factors suggested to effect risk for EOC. Furthermore, the Menopause Rating Scale (MRS)[18] was added.

MRS is a validated questionnaire available in several languages, including Swedish. It has 11 questions on sweating, heart discomfort, sleep problems, depressive mood, irritability, anxiety, physical and mental exhaustion, sexual problems, bladder problems, vaginal dryness and joint and muscular function, to which patients respond in a 5-grade Likert scale.[19]

Perioperative variables in GynOp are type of anaesthesia, any pathological finding in the abdomen, procedure(s) performed, complications, use of antibiotics, operative time, route of specimen removal from the

abdomen, blood loss, type of suturing and codes for surgery. SALSTER-specific questions concern total number and size of trocars used, method for tubal ligation, type of devices applied for salpingectomy, specific questions on method of specimen extraction and need to suture the muscle fasciae following specimen removal.

GynOp automatically sends questionnaires to the patients electronically at 8 weeks and 1 year postoperatively, to assess use of analgesics, bleeding, low urinary tract symptoms, sick leave, time to daily activities, satisfaction after surgery, complications and their treatment. If no answer is received, two digital reminders are sent automatically and thereafter by ordinary mail. Patient-reported complications are assessed and documented by a gynaecologist. Any complication is registered according to the Clavien-Dindo classification.[20] No amendments have been made to the 8-weeks questionnaire.

The 1-year questionnaire holds questions relating to pain experience, oestrogen treatment, symptoms from vagina, bladder and rectum, sexual intercourse last 3 months, coitus pain, result and satisfaction after surgery, complications, treatment of complications, hospital care and sick leave. The questionnaire has been supplemented with trial-specific questions on oestrogen and/or progesterone hormonal treatments and their indication, MRS, menstruation pattern, unintended pregnancies and their outcomes and smoking habits.

Routinely there is no further follow-up from GynOp. For trial participants questionnaires are sent every other year until the age of 55. Questions relate to the use of menopausal hormone therapy (MHT) or oestrogen and/ or progesterone hormonal treatments and their indication, MRS, bleeding pattern, smoking habits and unintended pregnancies and their outcomes.

### Eligibility

All patients planned for laparoscopic sterilisation are automatically screened for eligibility in the trial by the GynOp software. Patients with a known hereditary susceptibility for EOC such as BRCA (BReast CAncer) gene mutations are not considered for tubal ligation and thus not for inclusion in SALSTER. Potential trial participants can read online the SALSTER information and answer the specific study questions. Paper printouts are also available in which case a medical administrator registers the information in GynOp by using a login with a two-factor authentication system. Patients may also be informed about the trial at an outpatient clinical visit when the decision on sterilisation is taken. Informed consent (online supplemental appendix 1) can be given, usually online within GynOp or by signing a paper document at any time point before randomisation. The consent is kept safe according to established research routines. Inclusion and exclusion criteria are summarised in table 1.

### Randomisation and blinding

The randomisation module in GynOp randomly allocates women in proportion 1:1 to either salpingectomy or

**Table 1** Eligibility criteria for women participating in SALSTER

| Inclusion criteria | Exclusion criteria |
|---|---|
| ► Scheduled for laparoscopic sterilisation. <br> ► Willing to be randomised. | ► Women older than 49 years. <br> ► Not able to understand oral or written study information. <br> ► Previously treated for malignancy with either chemotherapy, radiotherapy or hormonal therapy which may negatively affect ovarian function. |

tubal ligation using permuted blocks with random sizes of either two or four while stratified for centre. Timing of randomisation is as close as possible to the time of surgery. The randomisation is performed online by the examining/operating gynaecologist or assistant with an immediate allocation response.

The nature of the trial makes blinding of patients very difficult and impossible for surgeons. Our intention is to avoid revealing information about which type of surgery was performed and we ask trial participants not to read their online medical records. However, the right to read medical records is regulated by law. Blinding of patients is further aggravated as detailed preoperative information is given including the number of scars associated with each procedure. In general, tubal ligation requires only one accessory port whereas salpingectomy requires at least two. Hence, blinding is not guaranteed.

### Interventions

Both interventions are planned as laparoscopic procedures. If the allocated procedure cannot be executed because of either unexpected pathology or high risk for serious intraoperative complications, the surgical procedure that was eventually performed will be registered in GynOp, but the individual still contributes with follow-up data. The same applies if extra surgical procedures are needed or in case of conversion to laparotomy where all surgical interventions are registered.

### Follow-up

Hospital staff routinely register data in GynOp at the end of every surgical procedure and at discharge. In case of a complication the surgeon registers the event. Responsible surgeon assesses the 8-weeks and 1-year questionnaires and in suspicion of a complication or unsatisfactory surgical results, a consultation is arranged. Any adverse effect is registered in GynOp. If there is no response after two routine reminders a member of the steering group contacts the department. In every department, a responsible physician will check responses and completeness of questionnaires at different time points. In case of an adverse event, any need for medical treatment to trial participants is covered by the Swedish healthcare system according to the Swedish law.

**Table 2** Outcomes in SALSTER

| Time interval | Primary outcomes | Secondary outcomes |
|---|---|---|
| Short-term (up to 8 weeks) | ► Any complication. | ► Severe complications.<br>► Operative time.<br>► Perioperative blood loss.<br>► Length of hospital stay. |
| Intermediate term (1 year after surgery) | | ► Complications according to Clavien-Dindo.<br>► Complications according to the existing questions on complications in GynOp. |
| Intermediate and long-term | | ► Subsequent surgery on uterus, salpinges and/or ovaries.<br>► Pregnancy rate. |
| Long-term (more than 1 year and up to 30 years after surgery) | ► Age at onset of natural menopause. | ► Age at the start of the perimenopausal state.<br>► Length of the perimenopausal state.<br>► Change in menopausal symptom score.<br>► Use of menopausal hormone therapy at any time during follow-up.<br>► Secondary expressions of oestrogen deficiency.<br>► Epithelial ovarian cancer. |

## Outcomes

The trial has two primary outcomes, one in the short-term and one in the long-term. Secondary outcomes are registered in the short-term, intermediate-term and long-term (table 2).

*Any complication* up to 8 weeks postoperatively, is retrieved directly from the GynOp database. The outcome includes any complication occurring perioperatively, diagnosed at postoperative emergency visits or noted by the patient and assessed by the physician in the 8-weeks questionnaire. The complication is further categorised as mild or severe, by organ damage and is graded according to the Clavien-Dindo classification. These categorised variables will be analysed as secondary outcomes.

*Age at onset of natural menopause*, defined as 12 months of amenorrhoea, is assessed by analysing reported bleeding pattern in the study-specific questionnaires sent every other year. Women with MHT prescription, oestrogen and/or progesterone hormonal treatments or a subsequent hysterectomy will not be included in this primary outcome, since they do not have a natural menopause.

The secondary short-term outcomes relate to the surgery and the in-hospital care as registered in GynOp. Secondary intermediate-term outcomes are retrieved from GynOp and other national quality and health registers. Secondary long-term outcomes such as length of and age at the start of perimenopausal state will be assessed by the trial-specific questionnaires describing bleeding pattern. Need for MHT will be assessed by every-other-year questionnaires and through The Drug Prescription Register up to 30 years after surgery. Uterine and adnexal surgery that occurs after the primary surgery will be assessed through GynOp at 1 year and The Patient register lifelong after surgery. Unintended pregnancies and their outcomes will be registered through the trial-specific questionnaires. If outcomes on ovarian function show a difference between groups, consequences of oestrogen deficiency, that is, fractures related to osteoporosis and cardiovascular events will be assessed through The Patient register.

Ovarian cancer will be assessed by cross-linking SALSTER with Swedish national registers and pooled with data from the ongoing Hysterectomy and OPPortunistic SAlpingectomy (HOPPSA) trial. HOPPSA is a Swedish multicentre, register-based RCT where patients planned for hysterectomy are randomised to salpingectomy or no salpingectomy.[21] By pooling data from SALSTER and HOPPSA the effect size of opportunistic salpingectomy to reduce the incidence of EOC will be estimated. Data will be retrieved through The Swedish Cancer Register, The Swedish Quality Register for Gynecological Cancer, The Swedish Cause of Death Register and The Swedish Population Register and at lifelong follow-up.

## Data monitoring and data management

Each surgical procedure in GynOp automatically receives a unique identification code number. This number is used in the trial to assign individual data, thus protecting confidentiality. The number of individuals randomised in the trial is continuously monitored by the GynOp's administrators. Numbers of recruited and percentage of eligible women per participating clinic are reported every 3 months on the GynOp website and through the SNAKS network which enhances communication between the research group and the departments participating in the trial. Regular online meetings are being held updating departments on the progress of the trial, and information is shared on recruiting performance. An independent appointed Data Safety Monitoring Board has performed an interim analysis when 50% of the target sample size was reached, according to the original plan and gave clearance for the study to continue recruiting patients.

## Patient and public involvement

Women in reproductive age in the general population were involved at an early phase of the planning, regarding choice of outcomes and development of the written study information. A short explanation of the research question and the intended study protocol in lay language with suggested outcomes were distributed among volunteers

in waiting rooms at gynaecology departments in Sweden. Open and specific questions were asked concerning the relevance of the trial, the design, the outcomes, any missing issues or missing outcomes. Questions associated with the draft of the written study information related to readability, unnecessary or missing information. Women were also asked to rate the importance of receiving information about potential risks associated with opportunistic salpingectomy.

## Statistics

### Sample size calculations

#### Primary short-term outcome: any complication up to 8 weeks

Complications to laparoscopic tubal ligation were registered in GynOp at a rate of 13.6% from 2010 to 2017. An increase of 3% is estimated after salpingectomy. If the non-inferiority margin is defined as +10%, the upper limit of the two-sided 95% CI ($\alpha$=0.05) for the difference between the salpingectomy and the tubal ligation groups shall not be above the +10% with a probability of 80% ($\beta$=0.20). To demonstrate non-inferiority, 411 women per randomisation group are needed (based on a two-sided Farrington-Manning test).[22] For protection against a 10% loss to follow-up, the target sample was determined at 914. The interim analysis revealed that 5% of randomised women interrupted their participation. For protection against this loss, the target sample size was increased to 968.

#### Primary long-term outcome: age at onset of menopause

Age at menopause on a Swedish population level was reported to be in mean 51.5 years and SD was estimated at 3.0. A decrease of 1 year is estimated after salpingectomy. If the non-inferiority margin is defined as 2 years, the upper limit of the two-sided 95% CI ($\alpha$=0.05) for the difference between the salpingectomy group and the tubal ligation group shall not be above 2 years with a probability of 80% ($\beta$=0.20). To demonstrate non-inferiority, 143 women per randomisation group are needed (two-sided non-parametric permutation test for comparison of two means). Considering exclusion of women without a natural menopause (30%), 5% of randomised women interrupting participation before the 8-weeks questionnaire and 15% loss during the 20 years long follow-up, approximately 572 women are needed for recruitment.

### Statistical plan

Both 'intention to treat', and 'per protocol' analyses will be performed. For non-inferiority design, the 'per protocol' analysis will be the primary.

*Any complication* will be presented as numbers along with percentages with 95% CI and the *age at onset of menopause* will be presented as mean and SD, as well as with median and quartiles. The two primary analyses measure different outcomes at different time points and will be published in separate articles. As they also test two different hypotheses, we will refrain from adjusting the 5% significance level for multiplicity.

## Analyses of any complication up to 8 weeks postoperatively

Primary analysis: To account for the lack of independence introduced by the stratification of the randomisation, we will estimate the difference in the complication risk between the two randomised groups with a 95% CI using a generalised estimation equation (GEE) with logistic link function, marginalised over centre and adjusted for age. The 95% CI of the risk difference will be estimated from the GEE-model using the delta method. The upper limit of the 95% CI shall not exceed the non-inferiority margin of 10%. As a sensitivity analysis, the unadjusted 95% CI for the difference in complications will be calculated according to Farrington-Manning.[22] Furthermore, unadjusted risk ratio (RR) and adjusted RR with 95% CI will also be calculated in secondary analyses using a GEE Poisson model with robust SEs.

## Analyses of age at menopause

The primary analysis will be a mixed effect model with adjustment for age as fixed effect and centre as random effect, from which a two-sided 95% CI for the mean difference will be constructed. The upper limit of the 95% CI shall not exceed the non-inferiority margin of 2 years for non-inferiority to be established. A sensitivity analysis without adjustment will be conducted by constructing a 95% CI for the mean difference using Fisher's non-parametric permutation test.

Missing data on the primary outcomes will be replaced with multiple imputation using fully conditional specification in the main analysis. In addition, a complete case analysis will be conducted. If both analyses of the two primary outcomes demonstrate non-inferiority, a common conclusion on the safety of the intervention can be inferred. However, the long period between these analyses will entail separate conclusions on complications and age at menopause, in a temporal order.

For other unadjusted comparisons between the two randomised groups Fisher's non-parametric permutation test will be used for continuous variables, Mantel-Haenszel $\chi^2$ test for ordered categorical variables, Fisher's exact test for dichotomous variables and $\chi^2$ test for non-ordered categorical variables. For dichotomous outcomes, a two-sided 95% CI for the difference in proportions between groups will be calculated as well as RRs with 95% CI. For continuous outcomes, two-sided 95% CIs for the difference in means between groups will be calculated. Also, adjusted analyses will be conducted.

All results from the secondary analysis will be given with estimates, 95% CI and two-sided p values, as well as unadjusted and adjusted RR with 95% CI. The analyses of the secondary endpoints will be mainly explanatory.

A detailed statistical analysis plan (SAP) will be written before data retrieval and published at the trial's site at ClinicalTrials.gov. Updates and changes in the planned statistical analyses will be published there.

### Nested trial of anti-Müllerian hormone levels

A biochemical measure of ovarian function is the serum level of AMH, a product of granulosa cells of the preantral and small antral follicles in the ovaries.[23] There is a theoretical rationale that salpingectomy may disturb the vascular and nervous supply to the ovary, or disrupt paracrine signalling, possibly causing impairment in ovarian function.[8] In the main trial, the primary outcome for ovarian function is based on clinical symptoms related to menopause. To strengthen the hypothesis of non-inferiority for ovarian function if salpingectomy is performed, an analysis of AMH is planned in a subset of patients.

Consecutive patients in SALSTER are asked for blood samples. Specific written and oral information is provided, and informed consent is signed. Blood samples are drawn at baseline and after 1 year. Seven hospitals are engaged in this nested trial. Samples are handled according to laboratory instructions, centrifugated, frozen within 2 days and stored in a biobank for later analysis, when the entire cohort will be analysed at the same time.

Results will be available after 1 year of follow-up and added manually to the GynOp data set. Patients wishing to be informed about their AMH levels result will be contacted. AMH levels will be compared between the salpingectomy versus tubal ligation groups and presented both in absolute and relative measures. Primary endpoint is absolute change in AMH from baseline to 1 year after surgery.

If non-inferiority is defined as 0.2 mg/L AMH, the upper limit of the two-sided 95% CI for the difference in change between the two groups shall not exceed 0.2 (SD for change 0.45) with a probability of 80% ($\beta=0.20$), and an estimation of up to 0.0 larger change (no difference in change) in the salpingectomy group, 81 patients per randomisation group is needed to show non-inferiority. Estimating a 20% loss to follow-up (a second blood sample not taken), 204 patients will be recruited in this nested trial. A two-sided 95% CI for the mean difference in absolute change in AMH will be constructed using a mixed effect model with adjustment for age as fixed effect and centre as random effect. Fisher's non-parametric permutation test will be applied for the unadjusted analysis.

## ETHICS AND DISSEMINATION

Even though EOC is not the most common gynaecological cancer it carries the worst prognosis due to early spread and vague symptomatology, making diagnosis difficult at an early stage. Based on the theory that the most common and aggressive form, HGSC may arise from the epithelium of the fallopian tubes, the practice of opportunistic salpingectomy has rapidly gained popularity. Well-designed trials have not been performed to study the safety profile of salpingectomy compared with tubal ligation regarding complications and the effect on ovarian function. SALSTER will assess if salpingectomy is as safe as tubal ligation. The withdrawal of hysteroscopic sterilisation made the trial ethically reasonable to design since the less invasive hysteroscopic procedure for sterilisation was not available anymore.[13] Regardless of the result, the trial will provide gynaecologists with high quality evidence to inform women, who can decide on having their tubes removed or not. If no additional risk is found, salpingectomy can be a recommended option. If not, the risks and benefits should be considered when counselling women wishing permanent surgical sterilisation.

SALSTER does not have EOC as a primary outcome for several reasons: There is a parallel trial, HOPPSA, which has EOC as a long-term primary outcome. At inclusion, the patients in HOPPSA are older than those in SALSTER, which implies a shorter time-to-event than in SALSTER. Also, hysterectomy is a more frequent procedure than sterilisation in Sweden, implying faster recruitment to the target sample size. Thus, the HOPPSA trial is more suited to investigate and conclude on EOC as a primary outcome. Furthermore, the plan for SALSTER is to contribute data to be pooled with HOPPSA data for the evaluation of the effect of opportunistic salpingectomy on EOC. A combined SAP will be written for an individual participant data meta-analysis combining HOPPSA and SALSTER.

The results of this trial will be presented at national as well as international scientific congresses and several publications are planned in international scientific journals. All results will be presented on an aggregated level, without any possibility to identify individuals. SNAKS will help to spread the results of this trial to its network of gynaecological departments in Sweden. Updates of results will be presented at the annual meetings of the Swedish Society of Obstetrics and Gynecology.

The SALSTER trial was approved by the central ethical review board in Gothenburg, Sweden, on 18 June 2018 (Dnr. 316–18). The first patient was randomised on 4 April 2019. The trial is recruiting, and 864 women had been randomised on 31 August 2022.

**Contributors** AS initiated the trial, designed and drafted the first study protocol. AI engaged in the revision and editing of the protocol. AI and MP are the primary contact persons with the GynOp register. KS contributes with ovarian tumour biology experience. LM initiated the AMH nested trial. AS, AI, KS, MP and LM approved the study protocol. AS applied to the Swedish Ethical Review Authority. PL wrote the statistical plan. LM wrote the first draft of this manuscript which was revised by AS, AI, KS and MP. All committed authors approved the final version of this manuscript. Principal investigator: AS, contact email: annika.strandell@vgregion.se.

**Funding** This work was supported by the Swedish Cancer Society (21 1408 Pj), the Lena Wäppling foundation, the Swedish state under the agreement between the Swedish government and the county councils, the ALF-agreement (ALFGBG-965130 and ALF RegVB-969584), Umeå University and Center for clinical research, county of Värmland grant number (LIVFOU-929703). The funders were not involved in study design; collection, management, analysis and interpretation of data; writing of the report; and the decision to submit the report for publication.

**Competing interests** None declared.

**Patient and public involvement** Patients and/or the public were involved in the design, or conduct, or reporting, or dissemination plans of this research. Refer to the Methods section for further details.

**Patient consent for publication** Not applicable.

**Provenance and peer review** Not commissioned; externally peer reviewed.

**ORCID iD**
Leonidas Magarakis http://orcid.org/0000-0001-9623-1302

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
