## [Reviewer comments · BMJ Open]

ARTICLE DETAILS

TITLE (PROVISIONAL)	SALpingectomy for STERilisation (SALSTER); Study protocol for a Swedish multicentre register-based randomised controlled trial.
AUTHORS	Magarakis, Leonidas; Idahl, Annika; Sundfeldt, Karin; Liv, Per; Pålsson, Mathias; Strandell, Annika

VERSION 1 – REVIEW

REVIEWER	Yoshihara, Kosuke Niigata University Graduate School of Medical and Dental Sciences
REVIEW RETURNED	07-Jan-2023

GENERAL COMMENTS	The authors introduced the protocol of SALSTER and provided substantial information of this study. To improve the quality of this manuscript, the authors should consider some points as follow: Page 5. Please sort out the relationship between the SALSTER and HOPPSA trials, as the HOPPSA trial appears abruptly on page 5. Page 10. As for the eligibility, please describe how to deal with BRCA status in this study because BRCA1/2 mutation status is very important to discuss the EOC risk. Page 12 and Table 2. Please clarify the definition of "long term".
---

REVIEWER	Pereira, Joana Margarida Araújo Unidade Local de Saúde do Alto Minho EPE, Obstetrícia e Ginecologia
REVIEW RETURNED	04-Mar-2023

GENERAL COMMENTS	Well-designed study but with ethical conflicts. With current knowledge, proposing tubal ligation is a matter of difficult analysis from an ethical point of view. I don't see the benefit of publishing the research protocol for the reader.
--

REVIEWER	Askary, Elham Shiraz University of Medical Sciences
REVIEW RETURNED	06-Jun-2023

GENERAL COMMENTS	It is better to write the whole proposal more concisely and avoid repeating phrases such as primary and secondary outcomes. I was not convinced about the blindness of the study. It is necessary to exclude patients who undergo laparotomy due to malignancy Avoid repeating the topics raised in the text again in the table.
---

VERSION 1 – AUTHOR RESPONSE

Reviewer 1:

Dr. Kosuke Yoshihara, Niigata University Graduate School of Medical and Dental Sciences

Comments to the Author:

The authors introduced the protocol of SALSTER and provided substantial information of this study. To improve the quality of this manuscript, the authors should consider some points as follow:

1. Page 5. Please sort out the relationship between the SALSTER and HOPPSA trials, as the HOPPSA trial appears abruptly on page 5.

Response from authors:

Thank you for commenting on this point. HOPPSA is a Swedish, multicentre, register-based randomised controlled trial of opportunistic salpingectomy at hysterectomy. One of its long-term objectives is to show that salpingectomy during hysterectomy is superior in reducing ovarian cancer incidence compared with hysterectomy alone.

Following your comment we have further explained the HOPPSA trial and moved the text to the Outcomes section (page 12, line 301-309).

2. Page 10. As for the eligibility, please describe how to deal with BRCA status in this study because BRCA1/2 mutation status is very important to discuss the EOC risk.

Response from authors:

We are grateful for this very important point. Women with any form of known hereditary susceptibility for ovarian cancer (BRCA or other mutations that increase the risk for ovarian cancer) follow specific guidelines for reducing ovarian cancer risk and are not planned for laparoscopic sterilisation. Consequently, they are not registered in the GynOp register and will not be screened for SALSTER. We have included a clarification regarding this topic (page 9, line 211-213).

3. Page 12 and Table 2. Please clarify the definition of "long term".

Response from authors:

We have added a comment in a parenthesis under the phrase 'long term' in Table 2 (page 11, line 266).

Reviewer 2:

Dr. Joana Margarida Araújo Pereira, Unidade Local de Saúde do Alto Minho EPE

Comments to the Author:

1, Well-designed study but with ethical conflicts. With current knowledge, proposing tubal ligation is a matter of difficult analysis from an ethical point of view.

Response from authors:

Thank you for your comment. As we do understand your point, our opinion is that more evidence is needed to support the recommendation of salpingectomy for sterilisation broadly. There are safety concerns with opportunistic salpingectomy at laparoscopy that are not yet addressed in RCTs, i.e. increased perioperative complications and negatively affected ovarian function, potentially contributing to an earlier menopause. For this reason, there was a call in 2016 for randomised trials to assess these issues. In our systematic literature review we found several knowledge gaps on the potential risks of opportunistic salpingectomy that needed to be addressed in order to correctly inform physicians and women.

At the same time salpingectomy to prevent epithelial ovarian cancer is supported by observational studies including patients with pathological tubes, and thus an indication for their removal leading to indication bias. The true effect size of opportunistic salpingectomy on EOC compared with no salpingectomy during gynaecological surgery is still unknown.

Altogether, the risk-benefit balance is still not known.

2. I don't see the benefit of publishing the research protocol for the reader.

Response from authors:

It is becoming more common to publish study protocols for RCTs, and it is a quality measure as the protocol cannot be changed to fit the results. It is also important for other researcher to find out about the study to relate their own research or planned trials.

Reviewer 3:

Dr. Elham Askary, Shiraz University of Medical Sciences

Comments to the Author:

1. It is better to write the whole proposal more concisely and avoid repeating phrases such as primary and secondary outcomes.

Response from authors:

We rephrased the section under General design, and believe that the proposal is concise now, with stated hypotheses regarding safety aspects, and with the addition that women with known hereditary risk for ovarian cancer will not be eligible. We moved the information about the HOPPSA trial (pages 5-6, lines 128-137). Repeating phrases have been overseen and deleted.

2. I was not convinced about the blindness of the study.

Response from authors:

We fully agree on this matter. The nature of the trial makes blinding of patients very difficult as the patients can read their medical records on-line, a right that is regulated by the Swedish law. The intention to blind is explained to trial participants and they are asked not to read their on-line medical records until they have responded to the one-year questionnaire. It is uncertain to what extent this advice is followed, and blinding is thus unpredictable. We have clarified that blinding is not guaranteed (page 9-10, line 223-227).

3. It is necessary to exclude patients who undergo laparotomy due to malignancy

Response from authors:

The trial only includes women planned for permanent contraception by a laparoscopic sterilisation procedure. Thus, no patients in need for cancer treatment or women diagnosed with suspected malignancy will be considered for inclusion, since they are not planned for laparoscopic sterilisation.

4. Avoid repeating the topics raised in the text again in the table.

Response from authors:

We have revised the manuscript and made a great effort not to be repetitive in the manuscript text and tables. We tried to formulate the written text as clarifications to the information in the tables.

VERSION 2 – REVIEW

REVIEWER	Yoshihara, Kosuke Niigata University Graduate School of Medical and Dental Sciences
REVIEW RETURNED	12-Jul-2023
GENERAL COMMENTS	The authors have responded to the reviewers' comments properly.
REVIEWER	Askary, Elham Shiraz University of Medical Sciences

	I am reeviewing the article "salpingectomy for STERilisation (SALSTER); Study protocol for a Swedish multicentre register-based randomised controlled trial.'
REVIEW RETURNED	16-Aug-2023

GENERAL COMMENTS	The proposed problems have been corrected and are acceptable
--